# Prognostic Laboratory Parameters in Placental Abruption: A Retrospective Case-Control Study

**DOI:** 10.3390/jcm8040482

**Published:** 2019-04-09

**Authors:** Sophie Pils, Chiara Paternostro, Christine Bekos, Marlene Hager, Robin Ristl, Johannes Ott

**Affiliations:** 1Department of Obstetrics and Gynecology, Medical University of Vienna, Spitalgasse 23, 1090 Vienna, Austria; sophie.pils@meduniwien.ac.at (S.P.); n1242333@students.meduniwien.ac.at (C.P.); christine.bekos@meduniwien.ac.at (C.B.); marlene.hager@meduniwien.ac.at (M.H.); 2Section for Medical Statistics, Center for Medical Statistics, Informatics, and Intelligent Systems, Medical University of Vienna, Spitalgasse 23, 1090 Vienna, Austria; robin.ristl@meduniwien.ac.at

**Keywords:** placental abruption, C-reactive protein, hemoglobin, leukocytes, fibrinogen

## Abstract

To evaluate routine laboratory parameters in women with and without placental abruption (PA) and in controls, 417 women were included in this retrospective cohort study in a tertiary-care center. 118 women with PA (Group A: 54 without vaginal bleeding and Group B: 64 with bleeding), 130 women without either PA or vaginal bleeding throughout their pregnancy (Group C), 123 women with vaginal bleeding but without PA (Group D), and 46 healthy pregnant women who had undergone a control laboratory evaluation in the second/third trimester for history of previous cytomegalovirus (additional control group) were included. Hemoglobin, leukocytes, thrombocytes, C-reactive protein (CRP), and fibrinogen were obtained within 48 h before C-section and/or at the time of bleeding onset. Cases (Groups A and B) revealed higher CRP levels than controls (Groups C and D) after multivariate analysis in the sub-analyses of bleeding (0.56 mg/dL, interquartile range (IQR) 0.28–1.24 vs. 0.51 mg/dL, IQR 0.28–0.84; odds ratio (OR) 1.108, *p* = 0.006) and non-bleeding women (0.64 mg/dL, IQR 0.48–1.08 vs. 0.32 mg/dL, IQR 0.18–0.61; OR 7.454, *p* < 0.001). The non-bleeding cases (Group A) revealed significantly higher leukocyte (12.01 g/L, IQR 9.41–14.10 vs. 9.21 g/L, IQR 7.95–10.49; OR 1.378, 95% confidence interval (CI): 1.095–1.735; *p* = 0.006) and CRP levels (0.64 mg/dL, IQR 0.48–1.08 vs. 0.33 mg/dL, IQR 0.20–0.50; OR 7.942, 95% CI: 1.435–43.958; *p* = 0.018) than the additional control group. In cases, none of the laboratory parameters differed between women with and without bleeding. The significantly increased CRP levels found for women with PA and the lack of a difference in CRP between bleeding and non-bleeding cases point toward a chronic process underlying placental abruption. However, this laboratory parameter does not seem clinically relevant for distinguishing between women with and without placental abruption at this point in time.

## 1. Introduction

Placental abruption (PA) is defined as partial or complete placental detachment prior to delivery of the fetus [1]. It is associated with substantial increases in maternal and fetal morbidity and mortality [2]. However, it is a rare condition, with an incidence of about one per cent [1]. As placental abruption is seen as ischemic placental disease, two pathways have been described: an inadequate placentation and an excessive detachment [3,4]. Hence, some risk factors are associated with a higher incidence, such as increased maternal age, conception by in vitro fertilization (IVF) [5], hypertension, smoking, trauma, uterine malformations, or drug abuse [6]; prediction of placental abruption, which has already been the topic of several studies, remains a challenging issue [7,8,9,10]. Notably, evidence suggests that placental abruption can be seen as a chronic process. Vaginal bleeding in early pregnancy and histologic lesions of the placenta, the umbilical cord, and the membranes have been shown to be associated with an increased risk of placental abruption in later pregnancy. Remarkably, chronic inflammatory placental lesions have not been associated with an increased risk, even in the absence of early vaginal bleeding. Based on these data, it can be assumed that prolonged inflammation may be involved in placental abruption [11]. At least in women with recurrent placental abruption, maternal immunologic responses against male-specific, minor histocompatibility antigens have been claimed to play a pathophysiologic role [12].

Hypothetically, such an inflammation might be detectable in the blood using routine parameters, regardless of whether it has been associated with an infection or not. The literature on this topic is scarce. As early as in the first trimester, C-reactive protein (CRP) levels, chlamydia pneumoniae-/trachomatis-specific immunoglobulins G and A, or CHSP60 antibody frequencies, are not altered in women with subsequent placental abruption [13]. Fibrinogen, known to be elevated during inflammation [14], has also been tested in women with placental abruption. However, no control group was included and the focus was on the prediction of adverse maternal and fetal outcomes [15].

Thus, we aimed here to evaluate routine laboratory parameters, i.e., hemoglobin, leukocytes, thrombocytes, CRP, and fibrinogen, in women with and without placental abruption. The second main focus was on the predictability of placental abruption, using these laboratory parameters, in two groups of clinical interest: women with and without one of the most obvious signs of placental abruption, namely, vaginal bleeding. In detail, we focused on the following questions: (i) In the case of vaginal bleeding, could the risk for placental abruption be stratified using the above-mentioned laboratory parameters? And (ii) did these parameters differ between women who suffered from placental abruption without vaginal bleeding, a circumstance that would be likely detected by CTG alterations, from women who presented with neither present placental abruption nor vaginal bleeding?

## 2. Material and Methods

### 2.1. Patient Population and Study Design

In this retrospective, case-control study, we included 417 women who delivered at the Department of Feto-maternal Medicine of the Medical University of Vienna, Austria, from January 2003 to November 2016 and for whom the routine laboratory parameters were available within 48 h before delivery and/or at the occurrence of vaginal bleeding. The department is the national reference center for maternal-fetal medicine in eastern Austria and the annual number of deliveries was at least 2500 during the study period. 

In detail, in the study period, 128 women had been diagnosed with placental abruption. Ten had to be excluded, since no pre-delivery laboratory parameters were available. Thus, the case group consisted of 118 women with a placental abruption (Group A: 54 without vaginal bleeding and Group B: 64 with bleeding). Placental abruption was diagnosed when a retroplacental hematoma attached to the placenta was found on visual examination after delivery [15]. Therefore, only women delivered by C-section were included. One hundred and thirty women without either placental abruption or vaginal bleeding throughout their pregnancy (Group C), and who were delivered by C-section for either breech presentation, previous C-section, or according to the patient’s wish, served as controls. We enrolled only women who underwent C-sections as controls, since, in these cases, placental abruption could be retrospectively excluded in a more reliable manner than in women with a vaginal delivery. Furthermore, 123 women with vaginal bleeding without placental abruption (Group D) served as additional controls. Notably, it was impossible to match Groups B and D for gestational age at bleeding onset or delivery, either using 1:1 matching, or by propensity score matching, without excluding several cases. Hence, we refrained from matching for the whole study population. As an additional control group, all 46 healthy pregnant women who had undergone a control laboratory evaluation in the second and third trimester for history of cytomegalovirus infection in the first trimester within the study period were included. Thus, these women had undergone evaluation of all study-relevant laboratory parameters in the course of clinical routine and served as an additional healthy control group for non-bleeding cases with placental abruption. Notably, none of the included patients revealed any known additional diseases thought to have an impact on the evaluated laboratory parameters, which included liver diseases, previous thromboembolism, malignant diseases, and any other acute or chronic inflammatory processes either due to infections or autoimmune diseases.

The study was approved by the Institutional Review Board of the Medical University of Vienna (IRB number: 2090/2016) on August 1st 2017, and was valid for one year after approval. The study protocol was in accordance with the Helsinki Declaration and current Austrian law, and, thus, neither written nor verbal informed consent was necessary according to the Ethics Committee of the Medical University of Vienna. Therefore, it was not obtained. The data were de-identified for statistical analysis.

### 2.2. Parameters Analyzed

Relevant data were acquired retrospectively. The main outcome parameters were laboratory parameters, i.e., leukocytes and CRP as routine inflammation markers, hemoglobin and thrombocytes as markers for chronic or acute bleeding, and fibrinogen as a marker for both inflammation and bleeding, which had been collected from a peripheral vein within 48 h before delivery by C-section and/or at the time of bleeding onset in patients who presented with vaginal bleeding. Accordingly, in women with vaginal bleeding who did not undergo a C-section within 48 h after bleeding onset, two blood samples were available and were included in the study. In cases of acute placental abruption, blood samples had been taken directly before skin incision for emergency C-section. This had been the case for 65 women. All examined laboratory parameters had been determined in the ISO-certified central laboratory of the Vienna General Hospital, Austria. Reagents from Diagnostica Stago, France, were used for quantitative determination of fibrinogen by Clauss method. For the measurement of CRP levels, a latex-enhanced immunoturbidimetric test (Beckman Coulter, USA) was used. Blood counts were performed with fluorescence flow cytometry using XE-5000 (Sysmex, Kobe, Japan). Data on these parameters were collected using AKIM^®^ software (version 7, SAP Software Solutions Austria, Vienna, Austria). In addition, the following parameters were included: basic patient characteristics (gestational age at delivery, maternal age at delivery, maternal BMI, parity, and gestational diabetes) to make our cohort comparable to others; previously addressed risk factors for placental abruption (pregnancies after IVF, pregnancy-induced or preexisting hypertension, cigarette smoking during pregnancy, and presence of placenta previa) [5,6]; and the time interval between the vaginal bleeding and delivery in order to enable a better comparability between bleeding cases and controls. The latter data were gathered using the Viewpoint^®^ software (version 25.0, GE Healthcare, Wessling, Germany) which is the basic perinatologic database at the department.

### 2.3. Statistical Analysis

Nominal variables have been reported as numbers and frequencies, and continuous variables as medians and interquartile ranges. Nominal variables between groups were compared using the Chi square test, and binary regression analyses were applied for numerical variables. For subgroup analyses, univariate binary regression analyses were performed. Significant variables were entered into multivariate binary regression models. For these analyses, odds ratios, including the 95 per cent confidence interval (95% CI), *p*-values of the likelihood ratio tests, and areas under the ROC curves have been given. All analyses were performed using SPSS statistics for Windows, version 24.0 (SPSS Inc., Chicago, IL, USA), and *p*-values < 0.05 were considered statistically significant.

## 3. Results

### 3.1. Basic Comparison between Women with and without Placental Abruption

As a first step, basic patient characteristics were compared between women with placental abruption and the control group (Table 1). Patients of the case group were significantly older, suffered more often from pregnancy-induced/preexisting hypertension, and delivered significantly earlier, which was accompanied by lower neonatal weight. 

### 3.2. Prediction of Placental Abruption Using Routine Laboratory Parameters in Women with Vaginal Bleeding

In this subgroup analysis, we included only women with vaginal bleeding. There were 123 patients with vaginal bleeding but without placental abruption. In these women, the median time interval between the onset of the last bleeding episode before delivery and delivery was 24 days (IQR, 13–48) compared to 0 days (IQR 0–4) in women with both bleeding and placental abruption (*n* = 64; *p* < 0.001). In order to test the value of the laboratory parameters for the prediction of placental abruption in a clinically relevant manner, the laboratory parameters at the time of bleeding onset were included into the analysis. As demonstrated in Table 2, women with placenta abruption were younger, had conceived by in vitro fertilization more often, suffered from arterial hypertension during pregnancy more often, and had a placenta previa less frequently (*p* < 0.05). They had a higher gestational age at bleeding onset (median 32.57 weeks, IQR 26.43–35.00 vs. 29.14 weeks, IQR 26.29–32.86; *p* = 0.020), but delivered earlier (median 32.57 weeks, IQR 26.71–35.14 vs. 35 weeks, IQR 31.71–37.57; *p* < 0.001). When focusing on the laboratory parameters, only CRP levels were slightly but significantly increased in cases versus controls (0.56 mg/dL, IQR 0.28–1.24 vs. 0.51 mg/dL, IQR 0.28–0.84; *p* = 0.025). Since gestational age at delivery cannot be used as a parameter with which to predict placental abruption, it was not included in the multivariate predictive model, whereas all other univariately significant parameters were included. Notably, all included variables remained statistically significant in the multivariate binary regression model, which included CRP (OR 1.506, 95% CI: 1.071–2.117; *p* = 0.019). For this multivariate model, the area under the Receiver Operating Characteristic (ROC) curve was 0.921 (Figure 1A). 

### 3.3. Routine Laboratory Parameters’ Predictive Value for Placental Abruption in Women without Vaginal Bleeding

The second subgroup analysis comprised cases (*n* = 54) and controls (*n* = 130) without vaginal bleeding (Table 3). In the univariate analysis, placental abruption was significantly associated with lower maternal age, arterial hypertension, and higher gestational age at delivery, as well as higher neonatal weight and higher leukocyte, CRP, and fibrinogen serum levels. Again, a multivariate binary regression model was conducted. All significant parameters were included, apart from neonatal weight, since it was thought to be redundant with gestational age at delivery. In this analysis, the following parameters remained significantly predictive for placental abruption: lower maternal age (OR 0.897, 95% CI: 0.807–0.997; *p* = 0.043); lower gestational age at delivery (OR 0.869, 95% CI: 0.814–0.926; *p* < 0.001); and higher CRP levels (OR 7.454, 95% CI: 1.538–36.121; *p* = 0.013). The median levels of the latter were found to be twice as high as those in the controls (0.64 mg/dL, IQR 0.48–1.08 vs. 0.32 mg/dL, IQR 0.18–0.61). The corresponding area under the ROC curve was 0.907 (Figure 1B).

In addition, the group of non-bleeding women with placental abruption was compared to women who underwent a laboratory control due to cytomegalovirus infection in the first trimester and, thus, were considered completely otherwise healthy. In this analysis, the multivariate model revealed that significantly higher leukocyte (OR 1.378, 95% CI: 1.095–1.735; *p* = 0.006) and CRP levels (OR 7.942, 95% CI: 1.435–43.958; *p* = 0.018) were found for cases than for these healthy controls.

### 3.4. Women with Placental Abruption: Comparison between Bleeding and Non-Bleeding Patients 

In a final step, we focused on cases only and compared those who presented with vaginal bleeding (*n* = 64) to those who did not (*n* = 54; Table 4). As demonstrated in the univariate analyses, patients with bleeding were significantly older, had conceived via IVF more often, and delivered at a lower gestational age (*p* < 0.05). In the multivariate model, only the two latter parameters remained significant (OR 4.076, 95% CI: 1.132; 14.679; *p* = 0.032 and OR 0.983, 95% CI: 0.973; 0.992; *p* < 0.001, respectively), which did not hold true for any of the laboratory parameters that differed between the groups. The corresponding area under the ROC curve was 0.686 (Figure 1C).

## 4. Discussion

This retrospective case-control study revealed that placental abruption was associated with slightly but significantly increased CRP levels. When focusing on subgroup analyses (bleeders and non-bleeders), it became evident that the other tested parameters were of no or minor relevance. To the best of our knowledge, this is the first case-control study about CRP, hemoglobin, leukocyte, thrombocyte, and fibrinogen levels in placental abruption. Several study limitations are addressed below and mainly include the retrospective design and control group selection.

Notably, the concept of ischemic placental disease is based on two major factors, namely, inadequate placentation and an excessive detachment pathway [3,4]. In the analyses, we included placenta previa and inflammation markers, respectively. Increased vascular pressure due to either pre-existing or pregnancy-induced hypertension, which is thought to act as an additional influencing factor, was also included in the multivariate models. However, the total number of patients affected by these additional conditions was low. Thus, performance of additional sub-analyses did not seem promising and would have introduced additional statistical bias due to excessive multiple testing. We consider this a major study limitation, especially since simultaneous hypertension and conception via IVF increase the risk of placental abruption [16]. This subanalysis could not be performed due to the small number of patients. 

Since the major aim of our study was to evaluate the predictive value of the laboratory parameters, we feel that the multivariate models were sufficient. Evaluation of the outcome “placental abruption” by use of these concepts is warranted in future scientific work. Using multicenter biobanking studies would probably solve the problem of acquiring high quality data on such a rare pregnancy complication.

However, our data seem to be in conflict with previous literature with regard to two well-known risk factors for placental abruption, namely, increased maternal age and placenta previa [5,17]. In our study, the control group had an increased maternal age and a higher incidence of placenta previa. This is likely due to the selection of controls. First, the latter finding seems reasonable, since placenta previa is associated with more chronic bleeding, and was thus overrepresented in the control group with bleeding (Table 1 and Table 2). When excluding patients with vaginal bleeding but without placental abruption from the analysis in Table 1, placenta previa was significantly associated with placental abruption (*p* = 0.007). Without doubt, placenta previa is a risk factor for placental abruption [17]. Second, maternal age was higher in the control group, which consisted of women who had undergone C-sections electively or for history of a previous C-section. Women with such a history can be expected to be older.

One important factor that might have influenced the laboratory parameters was gestational age at the time of blood sampling. Gestational age at delivery was higher in controls in both the bleeding and the non-bleeding subgroup analyses. By contrast, vaginal bleeding began earlier in the controls. It seems worth mentioning that, for these parameters, but also for others such as maternal age and conception via IVF, neither case-control matching nor propensity score-matching was feasible. This led to the necessity of performing multivariate analyses to rule out a major impact of the unbalanced patient characteristics. We know that this approach is a study limitation. Notably, CRP levels slightly decreased throughout pregnancy in a significant manner [18]. Nonetheless, we assumed that the multivariate analyses (Table 2 and Table 3) should have overcome the influence of gestational age on laboratory parameters. In addition, we included a control group of women who had undergone laboratory evaluation due to cytomegalovirus infection during the first trimester and were considered healthy at the time of blood sampling. In this multivariate analysis (Table 5), the non-bleeding cases revealed significantly higher leukocyte and CRP levels. The latter supports the other findings on CRP in this report. 

The most important findings are the increased CRP serum levels in women with placental abruption compared to controls, which became evident in patients both with and without vaginal bleeding (Table 2 and Table 3). These results remained significant in the multivariate models, which included the strong risk factors of arterial hypertension and gestational age. Thus, CRP can be assumed to be an independent predictor for placental abruption. The difference in median CRP levels was higher in the analysis of non-bleeding patients (0.64 mg/dL vs. 0.32 mg/dL) than in the analysis of bleeding patients (0.56 mg/dL vs. 0.51 mg/dL). Empirically, bleeding is generally considered to induce CRP increases. However, data on this are scarce. We have found only one study about inflammatory markers and aneurysmal subarachnoid hemorrhage that proves this hypothesis [19].

Notably, the comparison between placental abruption cases with and without vaginal bleeding demonstrated that CRP, as well as all other parameters, did not differ between these two groups at the time of the last blood analysis before delivery (Table 4). It has already been mentioned that placental abruption was a chronic process, except for cases that were attributable to acute trauma. Ananth et al. observed an increased risk of abruption in cases of placental inflammatory lesions, which suggests that the pathophysiologic and etiologic basis for placental abruption is a chronic inflammatory processes [11]. In our data set, the increases in CRP levels observed in cases could be due to this slow development of a clinically recognizable status of placental abruption. On the one hand, a chronic sterile inflammatory process could induce the CRP alterations [19]. On the other hand, it could be caused by minor chronic bleeding beneath the placenta, slowly abrupting from the decidual zone. In this study, placental abruption was defined by the finding of a retroplacental hematoma attached to the placenta. Thus, there must have been some bleeding. Based on this, one could assume that, in women without vaginal bleeding, the mentioned minor, local bleeding was merely clinically unrecognizable. This might be linked to the phenomenon that in women without vaginal bleeding, cases revealed significantly lower fibrinogen, but higher leukocyte levels than the controls (Table 3), which could reflect the wastage of coagulation factors and a chronic inflammatory process, respectively. Although a rise in fibrinogen during healthy pregnancies has been described, this effect is attenuated in patients with hypertension [20,21]. Thus, lower fibrinogen levels can also be explained by the higher percentage of hypertension and the earlier pregnancy week in the placental abruption group.

The patients with and without vaginal bleeding, and with placental abruption, did not differ in terms of CRP, leukocytes, fibrinogen, hemoglobin, and thrombocytes close to delivery (Table 4). This might also show that the process that finally leads to placental abruption can be seen as chronic. If, for example, placental abruption would have been an acute event in bleeding women, whereas only in non-bleeding women the underlying process would have been chronic, one might expect higher CRP and leukocytes, as well as lower fibrinogen levels, in the latter patients. We know that this assumption is highly hypothetical without comparable literature, but it is worth addressing. Another hint is the fact that women who did not have vaginal bleeding underwent C-sections about one week later than those with vaginal bleeding. In other words, in women with both no vaginal bleeding and those patients with vaginal bleeding and with placental abruption, there likely might have been a chronic sub-placental bleeding process that could only be clinically recognized in the patient with vaginal bleeding. However, no information about the amount of bleeding was available, which was due to the retrospective study design. We consider this a study limitation. However, no differences in hemoglobin or thrombocyte counts were found in any of the subgroup analyses. Thus, it could be assumed that, in both groups, blood loss in the probably chronic process that led to final placental abruption was of minor or no relevance, or at least not clinically determinable. 

Last not least, we have to address the fact that we cannot provide data on additional inflammation markers such as procalcitonin or interleukin-6 as a study limitation since procalcitonin has a higher diagnostic accuracy than CRP [22] and interleukin-6 increases in early infection stages inducing CRP [23]. Evaluation of these parameters might be a promising issue for future studies on placental abruption.

## 5. Conclusions

The significantly increased CRP levels found for women with placental abruption, and the lack of a difference in CRP between bleeding and non-bleeding cases, seem in line with the theory of a chronic process being associated with placental abruption. However, this parameter does not seem clinically relevant for distinguishing between women with and without placental abruption for now. Further research should focus on the underlying pathophysiologic mechanisms. Moreover, longitudinal changes in inflammation markers in placental abruption might be of interest. These data might be derived from bio-banking studies, since the overall incidence of placental abruption is low.

## Figures and Tables

**Figure 1 jcm-08-00482-f001:**
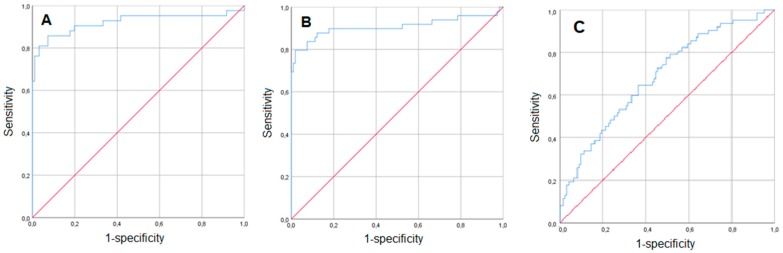
Prediction of placental abruption-ROC (Receiver Operating Characteristic) curves for the multivariate binary regression models presented in Table 2 (bleeding patients) (**A**), Table 3 (non-bleeding patients) (**B**), and Table 4 (patients with placental abruption) (**C**).

**Table 1 jcm-08-00482-t001:** Women with and without placental abruption: comparison of basic patient characteristics and laboratory parameters within 48 h before delivery between all cases and all controls.

		Placental Abruption (*n* = 118)	No Placental Abruption (*n* = 253)	OR (95% CI)	*p*-Value
Age (years) *		31.9 (26.3; 36.4)	33.65 (28.9; 36.8)	0.952 (0.917; 0.989)	0.011
Body mass index (kg/m^2^) *		23.4 (21.2; 26.2)	22.7 (20.2; 25.7)	1.046 (0.987; 1.110)	0.130
Pregnancy after IVF treatment ^#^		9 (7.6)	13 (5.1)	1.545 (0.633-3.673)	0.345
Parity ^#^	0	58 (48.7)	121 (48.0)	Reference	-
	1	28 (23.5)	75 (29.8)	0.779 (0.456–1.330)	0.360
	≥2	33 (27.7)	56 (22.2)	1.171 (0.686–1.998)	0.562
Pregnancy-induced/preexisting hyper-tension ^#^		21 (17.8)	8 (3.2)	6.630 (2.841–15.475)	<0.001
Smoking ^#^		25 (21.4)	39 (15.4)	1.491 (0.853–2.606)	0.159
Gestational diabetes mellitus ^#^		9 (7.6)	30 (11.9)	0.614 (0.282–1.338)	0.216
Placenta previa ^#^		14 (11.9)	53 (20.9)	0.508 (0.269; 0.958)	0.034
Neonatal weight (g) *		1839 (1076; 2500)	3044 (2400; 3430)	0.999 (0.999; 0.999)	<0.001
Gestational age at delivery (completed weeks) *		33.43 (28.86; 36.00)	38.14 (35.29; 38.86)	0.974 (0.966; 0.981)	<0.001
Leukocytes (g/L) *^+^		11.99 (9.82; 14.10)	10.07 (8.36; 12.05)	1.172 (1.091; 1.258)	<0.001
Thrombocytes (g/L) *^+^		218.5 (172; 267)	226 (184.5; 264.5)	0.998 (0.995; 1.001)	0.195
C-reactive protein (mg/dL) *^+^		0.58 (0.37; 1.14)	0.46 (0.24; 0.79)	1.093 (0.936; 1.276)	0.260
Fibrinogen (mg/dL) *^+^		437 (337; 519)	488 (431; 554)	0.994 (0.936; 1.276)	<0.001
Hemoglobin (g/dL) *^+^		11.20 (10.1–12.1)	11.70 (10.80–12.60)	1.009 (0.987–1.032)	0.423

Data are provided as * median and interquartile ranges or ^#^ numbers and frequencies; italic letters indicate statistical significance. ^+^ Laboratory parameters within 0–48 h before delivery. OR: Odds Ratio, IVF: in vitro fertilization.

**Table 2 jcm-08-00482-t002:** Women with and without placental abruption who presented with vaginal bleeding: comparison of basic patient characteristics and laboratory parameters at the time of bleeding onset.

		Placental Abruption (*n* = 64)	No Placental Abruption (*n* = 123)	OR (95% CI)	*p*-Value	OR (95% CI)	*p*-Value
Age (years) *		32.9 (28.5; 36.9)	33.6 (28.7; 36.5)	0.985 (0.932; 1.040)	0.581	-	-
Body mass index (kg/m^2^) *		23.15 (21.20; 29.05)	22.90 (20.20; 25.40)	1.067 (0.986; 1.155)	0.109	-	-
Pregnancy after IVF treatment ^#^		8 (12.5)	4 (3.3)	4.250 (1.228; 14.709)	0.014	5.594 (1.365; 22.915)	0.017
Parity ^#^	0	32 (50.0)	48 (39.0)	Reference	0.301	-	-
	1	15 (23.4)	40 (32.5)	1.373 (0.660; 2.854)	0.396	-	-
	≥2	17 (26.6)	35 (28.5)	0.772 (0.337; 1.769)	0.541	-	-
Pregnancy-induced/preexisting hypertension ^#^		8 (12.5)	1 (0.4)	17.429 (2.128; 142.72)	0.008	25.477 (2.347; 276:585)	0.008
Smoking ^#^		13 (20.6)	19 (15.4)	1.423 (0.651; 3.111)	0.375	-	-
Gestational diabetes mellitus ^#^		3 (4.7)	13 (10.6)	0.416 (0.114; 1.517)	0.172	-	-
Placenta previa ^#^		11 (17.2)	50 (40.7)	0.303 (0.144; 0.637)	0.001	0.219 (0.089; 0.538)	0.001
Neonatal weight (g) *		1665 (907; 2475)	2400 (1830; 2940)	0.999 (0.999; 1.000)	<0.001	-	-
Gestational age at bleeding onset (weeks) *		32.57 (26.43; 35.00)	29.14 (26.29; 32.86)	1.011 (1.002; 1.021)	0.020	1.108 (1.030; 1.192)	0.006
Gestational age at delivery (weeks) *		32.57 (26.71; 35.14)	35 (31.71; 37.57)	0.982 (0.973; 0.992)	<0.001	-	-
Hemoglobin (g/dL) *^+^		11.60 (10.60; 12.10)	11.30 (10.60; 12.00)	0.969 (0.776; 1.210)	0.782	-	-
Leukocytes (g/L) *^+^		10.91 (8.99; 13.50)	10.59 (8.92; 12.11)	1.080 (0.977; 1.217)	0.124	-	-
Thrombocytes (g/L) *^+^		217 (180; 276)	235 (187; 283)	0.996 (0.992; 1.001)	0.101	-	-
C-reactive protein (mg/dL) *^+^		0.56 (0.28; 1.24)	0.51 (0.28; 0.84)	1.469 (1.050; 2.005)	0.025	1.506 (1.071; 2.117)	0.019
Fibrinogen (mg/dL) *^+^		478.0 (435.5; 532.5)	457.0 (428.0; 528.0)	0.998 (0.994; 1.001)	0.200	-	-

Data are provided as * median and interquartile ranges or ^#^ numbers and frequencies; italic letters indicate statistical significance. ^+^ Laboratory parameters at the day of bleeding onset. OR: Odds Ratio, IVF: in vitro fertilization.

**Table 3 jcm-08-00482-t003:** Non-bleeding women with and without placental abruption: comparison of basic patient characteristics and laboratory parameters within 48 h before Caesarean delivery.

		Placental Abruption (*n* = 54)	No Placental Abruption (*n* = 130)	OR (95% CI)	*p*-Value	OR (95% CI)	*p*-Value
Age (years) *		29.75 (25.90; 35.50)	33.65 (28.90; 37.50)	0.923 (0.875; 0.974)	0.004	0.897 (0.807; 0.997)	0.043
Body mass index (kg/m^2^) *		23.50 (21.00; 25.90)	22.40 (20.20; 25.90)	1.016 (0.928; 1.112)	0.734	-	-
Pregnancy after IVF treatment ^#^		1 (1.9)	9 (6.9)	0.254 (0.031; 2.053)	0.199	-	-
Parity ^#^	0	26 (48.1)	73 (56.2)	Reference	0.525	-	-
	1	13 (24.1)	35 (26.9)	0.522 (0.236; 1.156)	0.109	-	-
	≥2	15 (27.8)	22 (16.9)	0.545 (0.218; 1.359)	0.545	-	-
Pregnancy-induced/preexisting hypertension ^#^		13 (24.1)	8 (6.2)	4.835 (1.872; 12.492)	<0.001	1.204 (0.123; 11.748)	0.873
Smoking ^#^		12 (22.2)	20 (15.4)	1.571 (0.707; 3.494)	0.265	-	-
Gestational diabetes mellitus ^#^		6 (11.1)	17 (13.1)	0.831 (0.309; 2.236)	0.711	-	-
Placenta previa ^#^		3 (5.6)	3 (2.3)	2.490 (0.486; 12.747)	0.259	-	-
Neonatal weight (g) *		1918 (1440; 2680)	3340 (3075; 3620)	0.997 (0.996; 0.998)	<0.001	-	-
Gestational age at delivery (weeks) *		34.00 (30.29–37.43)	38.57 (38.29; 39.00)	0.873 (0.830–0.918)	<0.001	0.869 (0.814; 0.926)	<0.001
Hemoglobin (g/dL) *^+^		11.30 (10.25; 12.00)	12.10 (11.30; 13.00)	1.013 (0.986; 1.041)	0.343	-	-
Leukocytes (g/L) *^+^		12.01 (9.41; 14.10)	9.54 (8.13; 10.97)	1.392 (1.214–1.595)	<0.001	1.124 (0.863; 1.464)	0.385
Thrombocytes (g/L) *^+^		215 (169.5; 259)	219 (184; 260)	0.998 (0.993–1.003)	0.398	-	-
C-reactive protein (mg/dL) *^+^		0.64 (0.48; 1.08)	0.32 (0.18; 0.61)	6.099 (2.381–15.624)	<0.001	7.454 (1.538; 36.121)	0.013
Fibrinogen (mg/dL) *^+^		418 (334; 534)	485 (442; 535)	0.994 (0.990–0.998)	0.002	1.003 (0.995; 1.012)	0.234

Data are provided as * median and interquartile ranges or ^#^ numbers and frequencies; italic letters indicate statistical significance. ^+^ Laboratory parameters within 48 h before delivery. OR: Odds Ratio, IVF: in vitro fertilization.

**Table 4 jcm-08-00482-t004:** Bleeding and non-bleeding women with placental abruption: comparison of basic patient characteristics and laboratory parameters within 48 h before Caesarean delivery.

		Vaginal Bleeding (*n* = 64)	No Vaginal Bleeding (*n* = 54)	OR (95% CI)	*p*-Value	OR (95% CI)	*p*-Value
Age (years) *		32.9 (28.5; 36.9)	29.8 (25.9; 35.5)	1.066 (1.001; 1.136)	0.046	0.989 (0.933; 1.049)	0.720
Body mass index (kg/m^2^) *		23.2 (21.2; 29.05)	23.5 (21.0; 25.9)	1.056 (0.955; 1.169)	0.288	-	-
Pregnancy after IVF treatment ^#^		8 (12.5)	1 (1.9)	7.571 (0.916; 62.608)	0.030	4.076 (1.132; 14.679)	0.032
Parity^#^	0	32 (50.0)	26 (48.1)	Reference	0.980	-	-
	1	15 (23.4)	13 (24.1)	1.086 (0.457; 2.582)	0.852	-	-
	≥2	17 (26.6)	15 (27.8)	1.018 (03.68; 2.841)	0.972	-	-
Pregnancy-induced/preexisting hypertension ^#^		8 (12.5)	13 (24.1)	0.451 (0.171; 1.187)	0.101	-	-
Smoking ^#^		13 (20.6)	12 (22.2)	0.910 (0.375; 2.206)	0.835	-	-
Gestational diabetes mellitus ^#^		3 (4.7)	6 (11.1)	0.393 (0.094; 1.655)	0.190	-	-
Placenta previa ^#^		11 (17.2)	3 (5.6)	3.528 (0.930; 13.384)	0.052	-	-
Neonatal weight (g) *		1665 (907; 2475)	1918 (1440; 2680)	1.000 (0.999; 1.000)	0.082	-	-
Gestational age at delivery (weeks) *		32.6 (26.7; 34.0)	34.0 (30.3; 37.1)	0.985 (0.974; 0.997)	0.011	0.983 (0.973; 0.992)	<0.001
Hemoglobin (g/dL) *^+^		11.2 (9.7; 12.1)	11.3 (10.25; 12.0)	0.879 (0.701; 1.103)	0.265	-	-
Leukocytes (g/L) *^+^		11.89 (9.89; 14.1)	12.00 (9.41; 14.10)	0.987 (0.889; 1.094)	0.799	-	-
Thrombocytes (g/L) *^+^		218.5 (175.0; 277.0)	215.0 (169.5; 259.0)	1.000 (0.995; 1.005)	0.900	-	-
C-reactive protein (mg/dL) *^+^		0.56 (0.28; 1.24)	0.64 (0.48; 1.08)	1.002 (0.773; 1.299)	0.985	-	-
Fibrinogen (mg/dL) * ^+^		440.5 (344.0; 494.5)	418 (334.0; 534.0)	1.000 (0.997; 1.003)	0.965	-	-

Data are provided as * median and interquartile ranges or ^#^ numbers and frequencies; italic letters indicate statistical significance. ^+^ Laboratory parameters within 48 h before delivery. OR: Odds Ratio, IVF: in vitro fertilization.

**Table 5 jcm-08-00482-t005:** Non-bleeding women with placental abruption and healthy patients after cytomegalovirus infection in the first trimester: comparison of basic patient characteristics and laboratory parameters.

		Placental Abruption (*n* = 54)	No Placental Abruption (*n* = 46)	OR (95% CI)	*p*-Value	OR (95% CI)	*p*-Value
Age (years) *		29.75 (25.90; 35.50)	30.55 (26,21; 33.70)	1.020 (0.951; 1.093)	0.588		
Body mass index (kg/m^2^) *		23.50 (21.00; 25.90)	22.40 (21.10; 26.70)	0.971 (0.885; 1.066)	0.540		
Pregnancy after IVF treatment ^#^		1 (1.9)	1 (2.2)	0.830 (0.050; 13.659)	0.896		
Parity ^#^	0	26 (48.1)	28 (60.9)	Reference	0.200		
	1	13 (24.1)	12 (26.1)	1.167 (0.452; 3.014)	0.750		
	≥2	15 (27.8)	6 (13.0)	2.692 (0.908; 7.983)	0.074		
Pregnancy-induced/preexisting hyper-tension ^#^		13 (24.1)	2 (4.3)	6.817 (1.448; 32.085)	0.015	4.565 (0.788; 26.433)	0.090
Smoking ^#^		12 (22.2)	2 (4.3)	6.143 (1.296; 29.121)	0.022	4.412 (0.778; 25.032)	0.094
Gestational diabetes mellitus ^#^		6 (11.1)	3 (6.5)	1.792 (0.422; 7.605)	1.792		
Placenta previa ^#^		3 (5.6)	0 (0)				
Gestational age at blood retrieval (weeks) *		34.00 (30.29–37.43)	33.86 (32.29; 37.00)	1.000 (0.984; 1.015)	0.953		
Hemoglobin (g/dL) *^+^		11.30 (10.25; 12.00)	11.55 (11.00; 12.10)	1.016 (0.965; 1.069)	0.542		
Leukocytes (g/L) *^+^		12.01 (9.41; 14.10)	9.21 (7.95; 10.49)	1.498 (1.229; 1.826)	<0.001	1.378 (1.095; 1.735)	0.006
Thrombocytes (g/L) *^+^		215 (169.5; 259)	217 (189; 261)	0.998 (0.991; 1.005)	0.600		
C-reactive protein (mg/dL) *^+^		0.64 (0.48; 1.08)	0.33 (0.20; 0.50)	20.849 (4.172; 104.182)	<0.001	7.942 (1.435; 43.958)	0.018
Fibrinogen (mg/dL) *^+^		418 (334; 534)	428 (387; 510)	1.000 (0.997; 1.003)	0.991		

Data are provided as * median and interquartile ranges or ^#^ numbers and frequencies; italic letters indicate statistical significance. ^+^ Laboratory parameters within 48 h before delivery. OR: Odds Ratio, IVF: in vitro fertilization.

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
