# Peer review of "Prognostic Laboratory Parameters in Placental Abruption: A Retrospective Case-Control Study"

_jcm, 2019, doi:10.3390/jcm8040482_

Round 1
Reviewer 1 Report
In this retrospective case-control study, Pils and colleagues aim to evaluate the routine laboratory parameters hemoglobin, leukocytes, thrombocytes, C-reactive protein (CRP) and fibrinogen in women with and without placental abruption in order to find a novel biomarker able to predict this disorder. Moreover, the Authors investigated whether placental abruption could be predicable in women with and without vaginal bleeding by using the above mentioned serum markers.
The main results of the present study was that placental abruption was associated with significantly increased CRP levels, while the other tested parameters were of no or minor relevance.
This is a well written paper that addresses the important issue of placental abruption prediction that still remains challenging and elusive.
Unfortunately, as also stated by the Authors, this work has several limitation as study design, control group selection and number of cases that do not allow to make definitive and clinically relevant conclusions from the data presented.
Author Response
Dear Editors,
Dear Reviewers,
We thank you for the comments and suggestions that helped us improve our manuscript. We took care in revising our manuscript according to these recommendations. We provide a marked manuscript as well as a point-by-point answer letter which you can find below. We hope that the revisions made to our manuscript will make it acceptable for publication in the “Journal of Clinical Medicine”. In case that there would be remaining queries, we shall be happy revising our work a second time.
Kind regards,
Johannes Ott
-- in the name of all authors --
Reviewer 1
In this retrospective case-control study, Pils and colleagues aim to evaluate the routine laboratory parameters hemoglobin, leukocytes, thrombocytes, C-reactive protein (CRP) and fibrinogen in women with and without placental abruption in order to find a novel biomarker able to predict this disorder. Moreover, the Authors investigated whether placental abruption could be predicable in women with and without vaginal bleeding by using the above mentioned serum markers.
The main results of the present study was that placental abruption was associated with significantly increased CRP levels, while the other tested parameters were of no or minor relevance.
This is a well written paper that addresses the important issue of placental abruption prediction that still remains challenging and elusive.
Unfortunately, as also stated by the Authors, this work has several limitation as study design, control group selection and number of cases that do not allow to make definitive and clinically relevant conclusions from the data presented.
Reply: We thank the reviewer for having assessed our manuscript. We agree with his worries about the study limitations. In accordance to the suggestions of Reviewer Number 3, we included a new control group in order to support our findings. Since we had to choose women without any additional health states that are known or could possible influence the evaluated laboratory parameters, we opted for women who had undergone a control laboratory evaluation in the second and third trimester for history of Cytomegalovirus infection in the first trimester. Despite the fact that matching was not possible, we believe that this control group is sufficient (see also Table 4: maternal age and gestational age at blood sampling did not differ between cases and controls). These women were compared to the non-bleeding cases. We hope that these new data improve the quality of presented data and strengthen the conclusion made.
In addition, we discuss the study limitations in the Discussion Section as follows:
“Several study limitations are addressed below and mainly include the retrospective design and control group selection.”
“[…] It seems worth mentioning that, for these parameters, but also for others such as maternal age and conception via IVF, neither case-control matching nor propensity score-matching was feasible. This led to the necessity of performing multivariate analyses to rule out a major impact of the unbalanced patient characteristics. We know that this approach is a study limitation.”
“[…]no information about the amount of bleeding was available, which was due to the retrospective study design. We consider this a study limitation.”
Methods: “As an additional control group, all 46 healthy pregnant women who had undergone a control laboratory evaluation in the second and third trimester for history of Cytomegalovirus infection in the first trimester within the study period were included. Thus, these women had undergone evaluation of all study-relevant laboratory parameters in the course of clinical routine and served as an additional healthy control group for non-bleeding cases with placental abruption”
Results: “In addition, the group of non-bleeding women with placental abruption was compared to women who underwent a laboratory control due to Cytomegalovirus infection in the first trimester and, thus, were considered completely otherwise healthy. In this analysis, the multivariate model revealed significantly higher leukocyte (OR 1.378, 95%CI: 1.095-1.735; p= 0.006) and CRP levels (OR 7.942, 95%CI: 1.435-43.958; p= 0.018) were found for cases than for these healthy controls.”
A new Table 4 has also been added to the manuscript.
Discussion: “In addition, we included a control group of women who had undergone laboratory evaluation due to Cytomegalovirus infection during the first trimester and were considered healthy at the time of blood sampling. In this multivariate analysis (Table 4), the non-bleeding cases revealed significantly higher leukocyte and CRP levels. The latter supports the other findings on CRP in this report.”
Reviewer 2 Report
Overall: Very interesting research. Improvements are suggested below:
introduction:
1. The introduction lacks a motivation on why the authors included parameters (line 99-104) which include "gestational age at delivery; the time interval between the vaginal bleeding and delivery; maternal age at delivery; maternal BMI; parity; pregnancies after in vitro fertilization (IVF); pregnancy-induced or preexisting hypertension; cigarette smoking during pregnancy; gestational diabetes; and presence of placenta previa" as mentioned in the method section.
2. The research questions need to be moved from the results to the introduction to make it easier for the reader to know what to expect in the discussion.
3. Be consistent in the use of terms: For instance, laboratory parameters, serum markers, routine markers and inflammation makers, inflammatory, laboratory markers. Do these mean the same thing or are different?
Discussion:
Most of the parameters are not explicitly discussed yet they have been mentioned in the introduction (51-52) as the key aspects which needed to be evaluated in the research. Currently, the discussion focuses mainly on the CRP and other parameters mentioned in point 1 above.
Grammar:
Lines 246-249 are not clear. Need to rephrase for better understanding.
Author Response
Dear Editors,
Dear Reviewers,
We thank you for the comments and suggestions that helped us improve our manuscript. We took care in revising our manuscript according to these recommendations. We provide a marked manuscript as well as a point-by-point answer letter which you can find below. We hope that the revisions made to our manuscript will make it acceptable for publication in the “Journal of Clinical Medicine”. In case that there would be remaining queries, we shall be happy revising our work a second time.
Kind regards,
Johannes Ott
-- in the name of all authors
Reviewer 2
introduction:
1. The introduction lacks a motivation on why the authors included parameters (line 99-104) which include "gestational age at delivery; the time interval between the vaginal bleeding and delivery; maternal age at delivery; maternal BMI; parity; pregnancies after in vitro fertilization (IVF); pregnancy-induced or preexisting hypertension; cigarette smoking during pregnancy; gestational diabetes; and presence of placenta previa" as mentioned in the method section.
Reply:
Concerning the parameters maternal age at delivery; parity; pregnancies after in vitro fertilization (IVF); pregnancy-induced or preexisting hypertension; cigarette smoking during pregnancy; and presence of placenta previa, the following revised sentence can be found in the manuscript: “However, it is a rare condition, with an incidence of about one per cent. Although several risk factors, such as increased maternal age, conception by in vitro fertilization (IVF), hypertension, smoking, trauma, uterine malformations, or drug abuse, have been described, prediction of placental abruption, which has already been the topic of several studies, remains a challenging issue .”
In order to keep the Introduction Section short, which is in accordance with the journal’s recommendations, we chose to provide a rationale for the other mentioned parameters in the Methods Section as follows: “In addition, the following parameters were included: basic patient characteristics (gestational age at delivery; maternal age at delivery; maternal BMI; parity; gestational diabetes) to make our cohort comparable to others; previously addressed risk factors for placental abruption (pregnancies after in vitro fertilization (IVF); pregnancy-induced or preexisting hypertension; cigarette smoking during pregnancy; presence of placenta previa); and the time interval between the vaginal bleeding and delivery in order to enable a better comparability between bleeding cases and controls.”
We hope that this is okay for the reviewer. If this issue should still remain a concern, we shall be happy revising the manuscript a second time.
2. The research questions need to be moved from the results to the introduction to make it easier for the reader to know what to expect in the discussion.
Reply: Thank you! The following sentences were moved from the Results to the Introduction Section: “In detail, we focused on the following questions: (i) In the case of vaginal bleeding, could the risk for placental abruption be stratified using the above-mentioned laboratory parameters? And (ii) did these parameters differ between women who suffered from placental abruption without vaginal bleeding, a circumstance that would be likely detected by CTG alterations, from women who presented with neither present placental abruption nor vaginal bleeding?”
3. Be consistent in the use of terms: For instance, laboratory parameters, serum markers, routine markers and inflammation makers, inflammatory, laboratory markers. Do these mean the same thing or are different?
Reply: All of these terms were changed to “laboratory parameters” throughout the manuscript.
Discussion:
Most of the parameters are not explicitly discussed yet they have been mentioned in the introduction (51-52) as the key aspects which needed to be evaluated in the research. Currently, the discussion focuses mainly on the CRP and other parameters mentioned in point 1 above.
Reply:
We agree. However, the Discussion Section needs to be kept short according to the journal’s recommendations. Thus, we chose to discuss only the most significant findings. Please find a discussion about these additional issues as follows in the revised Discussion Section:
“The patients with and without vaginal bleeding, and with placental abruption, did not differ in terms of CRP, leukocytes, fibrinogen, hemoglobin, and thrombocytes close to delivery (Table 5).”
“If, for example, placental abruption would have been an acute event in bleeding women, whereas only in non-bleeding women the underlying process would have been chronic, one might expect higher CRP and leukocytes, as well as lower fibrinogen levels, in the latter patients. We know that this assumption is highly hypothetical without comparable literature, but it is worth addressing.”
“Although a rise in fibrinogen during healthy pregnancies has been described, this effect is attenuated in patients with hypertension. So, the lower fibrinogen levels can also be explained by the higher percentage of hypertension and the earlier pregnancy week in the placental abruption group.“
“In addition, we included a control group of women who had undergone laboratory evaluation due to Cytomegalovirus infection during the first trimester and were considered healthy at the time of blood sampling. In this multivariate analysis (Table 4), the non-bleeding cases revealed significantly higher leukocyte and CRP levels. The latter supports the other findings on CRP in this report.”
“Although a rise in fibrinogen during healthy pregnancies has been described, this effect is attenuated in patients with hypertension [19,20]. Thus, the lower fibrinogen levels can also be explained by the higher percentage of hypertension and the earlier pregnancy week in the placental abruption group.”
Grammar:
Lines 246-249 are not clear. Need to rephrase for better understanding.
Reply: Rephrased as follows: “If, for example, placental abruption would have been an acute event in bleeding women, whereas only in non-bleeding women the underlying process would have been chronic, one might expect higher CRP and leukocytes, as well as lower fibrinogen levels, in the latter patients.”
Reviewer 3 Report
Dear Authors
This is a very interesting study, the problem of placental abruption is real as we still don't know much about it. However I have some major concerns about this study.
The inclusion and exclusion criteria are totally unknown. We don't know if the patients did not have any additional diseases or symptoms that might have an influence on the blood morphology or CRP.
There are many reference missing throughout the text. Many claims are unsupported by the relevant literature.
There is nothing about an idea of ischemic placental disease as the reason of placental abruption. As this is a major discussed concept I would edit the introduction and study outcomes with the use of that concept.
Some great literature.
https://www.sciencedirect.com/journal/seminars-in-perinatology/vol/38/issue/3
According to the above maybe it would be a good idea to compare different groups. Maybe authors should make an additional analysis on patiens with the hypertension problems alone. I I also think that we cannot make a group of patients with "hypertension" together as PIH and PPH are totally different ones.
Major concern. What about the gestational age during delivery? The main problem is that we cannot compare women e.g. in the late II trimester with the women in III trimester as they have totally different blood cell counts. There is a fluctuation of results during pregnancy, e.g. platelets https://onlinelibrary.wiley.com/doi/full/10.1002/ajh.24829
https://www.ncbi.nlm.nih.gov/pubmed/19935037 - fibrinogen levels are rising during pregnancy
What about matching the patients with placental abruption with healthy pregnant women of the same gestational age and the same age? The control group is easy to obtain in these cases.
Why only CRP? What about other inflammatory patterns like e.g. procalitonin, IL-6? Please describe. If there is some frozen blood serum, maybe the good idea would be to make some additional lab check.
Author Response
Dear Editors,
Dear Reviewers,
We thank you for the comments and suggestions that helped us improve our manuscript. We took care in revising our manuscript according to these recommendations. We provide a marked manuscript as well as a point-by-point answer letter which you can find below. We hope that the revisions made to our manuscript will make it acceptable for publication in the “Journal of Clinical Medicine”. In case that there would be remaining queries, we shall be happy revising our work a second time.
Kind regards,
Johannes Ott
-- in the name of all authors --
Reviewer 3
1. The inclusion and exclusion criteria are totally unknown. We don't know if the patients did not have any additional diseases or symptoms that might have an influence on the blood morphology or CRP.
Reply: We thank the reviewer for this comment. We forgot to include this essential information in the original manuscript and added the following to the revised version (Methods Section): “Notably, none of the included patients revealed any known additional diseases thought to have an impact on the evaluated laboratory parameters which included liver diseases, previous thromboembolism, malignant diseases, and any other acute or chronic inflammatory processes either due to infections of autoimmune diseases.”
2. There are many references missing throughout the text. Many claims are unsupported by the relevant literature.
Reply: We thank the reviewer. We included the following new references:
3. Parker SE, Werler MM. Epidemiology of ischemic placental disease: a focus on preterm gestations. Semin Perinatol. 2014;38(3):133-8.
4. Ananth CV. Ischemic placental disease: a unifying concept for preeclampsia, intrauterine growth restriction, and placental abruption. Semin Perinatol. 2014;38(3):131
10. Ananth CV, Vintzileos AM. Ischemic placental disease: epidemiology and risk factors. Eur J Obstet Gynecol Reprod Biol. 2011 Nov;159(1):77-82
19. Abbassi-Ghanavati M, Greer LG, Cunningham FG. Pregnancy and laboratory studies: a reference table for clinicians. Obstet Gynecol. 2009;114(6):1326-31.
20. Hale SA, Sobel B, Benvenuto A, Schonberg A, Badger GJ, Bernstein IM. Coagulation and Fibrinolytic System Protein Profiles in Women with Normal Pregnancies and Pregnancies Complicated by Hypertension. Pregnancy Hypertens. 2012;2(2):152-157.
3. There is nothing about an idea of ischemic placental disease as the reason of placental abruption. As this is a major discussed concept I would edit the introduction and study outcomes with the use of that concept. / 4. Some great literature. https://www.sciencedirect.com/journal/seminars-in-perinatology/vol/38/issue/3 / 5. According to the above maybe it would be a good idea to compare different groups. Maybe authors should make an additional analysis on patients with the hypertension problems alone. I also think that we cannot make a group of patients with "hypertension" together as PIH and PPH are totally different ones.
Reply: We thank the reviewer for making us aware of this interesting literature and concepts! We reply to all of these concerns at once, since the queries belong together. The concept of ischemic placental disease is based on two major factors, namely inadequate placentation and an excessive detachment pathway. In the analyses, we did include placenta previa for the first factor and inflammation markers for the second. Increased vascular pressure, an additional influencing factor, due to either pre-existing or pregnancy-induced hypertension, was also included into the multivariate models. However, the total numbers of patients affected by these additional conditions were low. Thus, performance of additional sub-analyses did not seem promising and would have introduced additional statistical bias due to excessive multiple testing. Since the major aim of our study was to evaluate the predictive value of the laboratory parameters, we feel that the multivariate models were sufficient. Notably, the relevance of infections could not be included into our model due to the retrospective design. We hope that the additional paragraph in the Discussion Section is sufficient for the reviewer. If we are recommended to perform additional revisions, we shall be happy doing so in future revisions.
We discuss this issue as follows in the Discussion Section and also include some of the relevant references: “Notably, the concept of ischemic placental disease is based on two major factors, namely inadequate placentation and an excessive detachment pathway. In the analyses, we include placenta previa and inflammation markers, respectively. Increased vascular pressure due to either pre-existing or pregnancy-induced hypertension, which is thought to act as an additional influencing factor, was also included into the multivariate models. However, the total number of patients affected by these additional conditions were low. Thus, performance of additional sub-analyses did not seem promising and would have introduced additional statistical bias due to excessive multiple testing. We consider this a major study limitation. Since the major aim of our study was to evaluate the predictive value of the laboratory parameters, we feel that the multivariate models were sufficient. Evaluation of the outcome “placental abruption” by use of these concepts is warranted in future scientific work. Probably, multicenter biobanking studies would solve the problem of acquiring high quality data on such a rare pregnancy complication.”
In addition, we added the following paragraph to the Introduction Section: “As placental abruption is seen as ischemic placental disease, two pathways were described: an inadequate placentation and an excessive detachment. Therefore some risk factors are associated with a higher incidence”
6. Major concern. What about the gestational age during delivery? The main problem is that we cannot compare women e.g. in the late II trimester with the women in III trimester as they have totally different blood cell counts. There is a fluctuation of results during pregnancy, e.g. platelets https://onlinelibrary.wiley.com/doi/full/10.1002/ajh.24829
Reply: We thank the reviewer for this comment. It is discussed as follows: “One important factor that might have influenced the laboratory results parameters was gestational age at the time of blood sampling. Gestational age at delivery was higher in controls in both the bleeding and the non-bleeding subgroup analyses. In contrast, vaginal bleeding began earlier in controls. It seems worth mentioning that, for these parameters, but also for others such as maternal age and conception via IVF, neither case-control matching nor propensity score-matching was feasible. This led to the necessity of performing multivariate analyses to rule out a major impact of the unbalanced patient characteristics. We know that this approach is a study limitation. Notably, CRP levels slightly decreased throughout pregnancy in a significant manner. Nonetheless, we assume that the multivariate analyses (Tables 2 and 3) should have overcome the influence of gestational age on laboratory parameters.”
Moreover, we included a new healthy control group and revised the Methods, Results, and Discussion Sections accordingly (see below).
7. https://www.ncbi.nlm.nih.gov/pubmed/19935037 - fibrinogen levels are rising during pregnancy
Reply: We included the new reference and a comment on fibrinogen in the Discussion Section:
“Although a rise in fibrinogen during healthy pregnancies has been described, this effect is attenuated in patients with hypertension. Thus, the lower fibrinogen levels can also be explained by the higher percentage of hypertension and the earlier pregnancy week in the placental abruption group.”
8. What about matching the patients with placental abruption with healthy pregnant women of the same gestational age and the same age? The control group is easy to obtain in these cases.
Reply: We thank the reviewer for this important comment. We included a new control group in order to support our findings. Since we had to choose women without any additional health states that are known to or could possible influence the evaluated laboratory parameters, we opted for women who had undergone a control laboratory evaluation in the second and third trimester for history of Cytomegalovirus infection the first trimester. Despite the fact that matching was not possible, we believe that this control group is sufficient (see also new Table 4: maternal age and gestational age at blood sampling did not differ between cases and controls). These women were compared to the non-bleeding cases. Please find the following new paragraphs in the revised manuscript:
- Methods: “As an additional control group, all 46 healthy pregnant women who had undergone a control laboratory evaluation in the second and third trimester for history of Cytomegalovirus infection the first trimester within the study period were included. Thus, these women had undergone evaluation of all study-relevant laboratory parameters in the course of clinical routine and served as an additional healthy control group for non-bleeding cases with placental abruption”
- Results: “In addition, the group of non-bleeding women with placental abruption was compared to women who underwent a laboratory control due to Cytomegalovirus infection in the first trimester and, thus, were considered completely otherwise healthy. In this analysis, the multivariate model revealed significantly higher leukocyte (OR 1.378, 95%CI: 1.095-1.735; p= 0.006) and CRP levels (OR 7.942, 95%CI: 1.435-43.958; p= 0.018) were found for cases than for these healthy controls.”
- A new Table 4 has also been added to the manuscript.
- Discussion: “In addition, we included a control group of women who had undergone laboratory evaluation due to Cytomegalovirus infection during the first trimester and were considered healthy at the time of blood sampling. In this multivariate analysis (Table 4), the non-bleeding cases revealed significantly higher leukocyte and CRP levels. The latter supports the other findings on CRP in this report.”
Why only CRP? What about other inflammatory patterns like e.g. procalitonin, IL-6? Please describe. If there is some frozen blood serum, maybe the good idea would be to make some additional lab check.
Reply: We are sorry to say that due to the retrospective design, these parameters were not evaluated. The majority of patients were not subject to biobanking, thus, no frozen blood serum was available. We added the following statement to the Discussion Section: “Last not least, we have to address the fact that we cannot provide data on additional inflammation markers such as procalcitonin or interleukin-6 as a study limitation.
Round 2
Reviewer 1 Report
The Authors improved the manuscript but unfortunately the study limitations are still present and the conclusions are not supported enough.
Author Response
Reviewer 1
The authors improved the manuscript but unfortunately the study limitations are still present and the conclusions are not supported enough.
Reply:
We thank the reviewer for his/her effort in re-evaluation of our manuscript. We agree on the issue about the study limitations. We do believe that these are discussed extensively in the Discussion Section.
Concerning the support of our conclusions: In the Abstract, we only state that “[…] this laboratory parameter does not seem clinically relevant for distinguishing between women with and without placental abruption at this point in time.” Our data clearly point out that CRP cannot be used for reliably distinguishing between women with and without placental abruption. Thus, we do not feel that this was unsupported. In our eyes, given the concept of ischemic placental disease, this finding seems clinically relevant and worth reporting.
However, we believe that the reviewer is worried about the conclusion that the increased CRP levels point towards an underlying chronic inflammatory process. Indeed, we agree. Thus, we rephrased this conclusion as follows: “However, the significantly increased CRP levels found for women with placental abruption, and the lack of a difference in CRP between bleeding and non-bleeding cases, seem in line with the theory of a chronic process being associated with placental abruption.”
Moreover, we want to put an emphasis on the fact that an additional control group has been added in the course of the last revision process which overcomes some of the previously mentioned study limitations.
We hope that the changes made to our manuscript by which we try to avoid any over-interpretation of our data will make it acceptable for the Reviewer.
Reviewer 3 Report
It was a pleasure to read the revised manuscript. I found it really interesting and these data might be useful for further studies in this area.
Main concerns
Line 13
I don’t know if authors have placed the correct numbers here – please check the groups and the numbers. Did you include here the new group?
Line 83
Why do you use both numbers and text? One hundred thirty might be changed to 130.
Table 2
What about a little discussion about the differences in IVFs and hypertension patients? What do you think about the results in placenta previa patients (aren’t they a group of higher risk of PA?)?
Line 354
I think it would be a good idea to add one or two sentences about the new ideas for further studies with the use of PCT or IL-6? Some idea about interleukins or growth factors, e.g TNF. Maybe some readers would use it in their studies.
Best regards
Author Response
Reviewer 3
Line 13
I don’t know if authors have placed the correct numbers here – please check the groups and the numbers. Did you include here the new group?
Reply: We are sorry for the mistake. We corrected the total number and included information about the additional control group to the Abstract:
- “To evaluate routine laboratory parameters in women with and without placental abruption (PA) and in controls, 417 women were included in this retrospective cohort study in a tertiary-care center. 118 women with PA (group A: 54 without vaginal bleeding, group B: 64 with bleeding) and 130 women without either PA or vaginal bleeding throughout their pregnancy (group C), 123 women with vaginal bleeding but without PA (group D), and 46 healthy pregnant women who had undergone a control laboratory evaluation in the second/third trimester for history of previous Cytomegalovirus (additional control group) were included.”
- “The non-bleeding cases (group A) revealed significantly higher leukocyte (12.01g/L, IQR 9.41-14.10 versus 9.21g/L, IQR 7.95-10.49; OR 1.378, 95%CI: 1.095-1.735; p= 0.006) and CRP levels (0.64mg/dL, IQR 0.48-1.08 versus 0.33 mg/dL, IQR 0.20-0.50; OR 7.942, 95%CI: 1.435-43.958; p= 0.018) than the additional control group.”
Line 83
Why do you use both numbers and text? One hundred thirty might be changed to 130.
Reply: The English native speaker who corrected the manuscript stated that this should be done in case of a number being the first word in a sentence. However, we corrected it to “130”. Thank you!
Table 2
What about a little discussion about the differences in IVFs and hypertension patients? What do you think about the results in placenta previa patients (aren’t they a group of higher risk of PA?)?
Reply: We thank the reviewer. The following paragraph had already been included in the Discussion Section: “However, our data seem to be in conflict with previous literature with regard to two well-known risk factors for placental abruption, namely, increased maternal age and placenta previa [5,17]. In our study, the control group had an increased maternal age and a higher incidence of placenta previa. This is likely due to the selection of controls. First, the latter finding seems reasonable, since placenta previa is associated with more chronic bleeding, and was, thus, overrepresented in the control group with bleeding (Table 1 and 2).”
In addition, we added the following new statements into the Discussion Section:
“[We consider this a major study limitation, especially since simultaneous hypertension and conception via IVF increase the risk of placental abruption [16]. This subanalysis could not be performed due to the small number of patients.”
“When excluding patients with vaginal bleeding but without placental abruption from the analysis in Table 1, placenta previa was significantly associated with placental abruption (p=0.007; data not shown). Without doubt, placenta previa is a risk factor for placental abruption [17].”
We also added the following new reference:
16. Dayan N, Lanes A, Walker MC, Spitzer KA, Laskin CA. Effect of chronic hypertension on assisted pregnancy outcomes: a population-based study in Ontario, Canada. Fertil Steril. 2016;105(4):1003-9.
Line 354
I think it would be a good idea to add one or two sentences about the new ideas for further studies with the use of PCT or IL-6? Some idea about interleukins or growth factors, e.g TNF. Maybe some readers would use it in their studies
Reply: We added the following lines to the Discussion Section:
“Last not least, we have to address the fact that we cannot provide data on additional inflammation markers such as procalcitonin or interleukin-6 as a study limitation since. Procalcitonin has a higher diagnostic accuracy than CRP [22] and Interleukin-6 increases in early infection stages inducing CRP [23]. Evaluation of these parameters might be a promising issue for future studies on placental abruption.”
Moreover, we included the following references:
22. Simon L, Gauvin F, Amre DK, Saint-Louis P, Lacroix J. Serum procalcitonin and C-reactive protein levels as markers of bacterial infection: a systematic review and meta-analysis. Clin Infect Dis. 2004;39(2):206-17.9
23. Tanaka T, Kishimoto T. The biology and medical implications of interleukin-6. Cancer Immunol Res. 2014;2(4):288-94.